# In Vitro Antitumor and Antioxidant Capacity as well as Ameliorative Effects of Fermented Kefir on Cyclophosphamide-Induced Toxicity on Cardiac and Hepatic Tissues in Rats

**DOI:** 10.3390/biomedicines12061199

**Published:** 2024-05-28

**Authors:** Songul Cetik Yildiz, Cemil Demir, Mustafa Cengiz, Halit Irmak, Betul Peker Cengiz, Adnan Ayhanci

**Affiliations:** 1Department of Medical Services and Techniques, Health Services Vocational School, Mardin Artuklu University, 47200 Mardin, Türkiye; cemildemir@ymail.com; 2Department of Elementary Education, Faculty of Education, Siirt University, 56100 Siirt, Türkiye; mustafacengizogu@gmail.com; 3Department of Computer Sciences, Mardin Artuklu University, 47200 Mardin, Türkiye; halitirmak@artuklu.edu.tr; 4Eskisehir Yunus Emre State Hospital, 26190 Eskisehir, Turkey; betip76@yahoo.com; 5Department of Biology, Science Faculty, Eskisehir Osmangazi University, 26040 Eskisehir, Türkiye; aayhanci@ogu.edu.tr

**Keywords:** kefir, cyclophosphamide, oxidative stress, antioxidant, antitumor, rats

## Abstract

Fermented prebiotic and probiotic products with kefir are very important to slow down and prevent the growth of tumors and to treat cancer by stimulating the immune response against tumor cells. Cyclophosphamide (CPx) is widely preferred in cancer treatment but its effectiveness in high doses is restricted because of its side effects. The aim of this study was to investigate the protective effects of kefir against CPx-induced heart and liver toxicity. In an experiment, 42 Wistar albino rats were divided into six treatment groups: the control (Group 1), the group receiving 150 mg/kg CPx (Group 2), the groups receiving 5 and 10 mg/kg kefir (Groups 3 and 4) and the groups receiving 5 and 10 mg/kg kefir + CPx (Group 5 and 6). Fermented kefirs obtained on different days by traditional methods were mixed and given by gavage for 12 days, while a single dose of CPx was administered intraperitoneally (i.p.) on the 12th day of the experiment. It was observed that alanine transaminase (ALT), aspartate transaminase (AST), alkaline phosphatase (ALP), lactate dehydrogenase (LDH), creatinine kinase-MB (CK-MB), ischemia modified albumin (IMA) and Troponin I values, which indicate oxidative stress, increased in the CPx-administered group, and this level approached that of the control in the CPx + kefir groups. Likewise, as a result of the kefir, the rats’ CPx-induced histopathological symptoms were reduced, and their heart and liver tissue were significantly improved. In conclusion, it was observed that kefir had a cytoprotective effect against CPx-induced oxidative stress, hepatotoxicity and cardiotoxicity, bringing their biochemical parameters closer to those of the control by suppressing oxidative stress and reducing tissue damage.

## 1. Introduction

The demand for probiotic and prebiotic products is increasing. Kefir, an important fermented product with these properties, is a foamy, thick, slightly sour drink that has been produced especially in the Caucasus region for many years and is now produced for commercial purposes in European and American countries [1]. Recently, studies have been carried out to reduce the side effects of chemotherapy drugs on healthy tissues and increase immunity without changing their antineoplastic effect. Many scientific studies have shown that antioxidants, which can strengthen the immune system and protect cell integrity, are agents that can alleviate or completely eliminate the side effects that may occur during the use of this drug. For this purpose, many studies have shown that antioxidants are useful in reducing the toxicity and oxidative damage caused by cyclophosphamide (CPx) [2,3,4]. Free radicals have the potential to seriously harm tissues’ DNA, proteins, carbohydrates, and lipids when their quantity exceeds the antioxidant capacity. Free radicals have harmful effects that lead to a variety of health issues, including cancer. The first stage of oxidative-stress-related carcinogenesis is constituted by free radicals [4,5]. Kefir has antioxidant and antitumor properties as well as immunotherapeutic and anticarcinogenic effects by slowing the growth of cancer cells and accelerating apoptosis. According to the results of research, kefir is a natural therapeutic agent for cancer [2]. The regular consumption of kefir can lead to antioxidant, anticarcinogenic, immune-system-regulating [6] and anticarcinogenic effects and can protect against apoptosis [6,7,8]. In addition to its antioxidative, antimicrobial and anticarcinogenic properties, kefir also shows anti-apoptotic activity. The anticarcinogenic effect of kefir is mostly due to the sulfur-containing amino acid groups in its structure. It has been stated that the microorganisms in kefir can significantly affect immune-regulating properties. It has also been stated that kefir can reduce lipid peroxidation and positively affect antioxidant parameters. Kefir accelerates apoptosis in cancerous cells by reducing mutation and DNA damage, decreasing the activities of enzymes (ß-glucuronidase, nitroreductase and azoreductase) that are predisposed to cancer formation, neutralizing mutagens, increasing the production of short-chain fatty acids and acidity and providing an anticarcinogenic effect. The lactic acid bacteria found in kefir have been shown to increase non-specific resistance to tumors or infections in humans and animals as well as enhance certain immune reactions. Immune activities have also been observed in various animals and humans after the consumption of lactic acid bacteria [2,9]. Despite its known hepatoprotective and cardioprotective [10] activities, there are no studies in the literature that report that kefir can protect liver and heart tissue from damage induced by the chemotherapy drug CPx.

CPx, which is widely used clinically, is a synthetic alkylating agent chemically related to nitrogen mustards. CPx must be given in high doses to prevent tumors from resisting anti-carcinogenic agents. However, while high doses prolong survival, they also increase possible toxic side effects [2]. CPx shows anticancer properties after being activated by hepatic cytochrome P450 enzymes. It functions as a tumor-blocking drug by inhibiting nucleic acid synthesis and stopping mitosis with its reactive metabolites, such as acrolein (ACR) and phosphoramide mustard (PM) [11,12]. PM, which is the antineoplastic part, becomes active when high doses of CPx are used, binds to DNA and then becomes alkylated, negatively affecting DNA transcription and replication [13]. ACR is a highly reactive aldehyde with α- and β-unsaturated groups and has been identified as an initiator of lipid peroxidation [14,15]. Oxidative stress and reactive oxygen species (ROS) are the main mechanisms involved in CPx-induced hepatotoxicity, causing the lipid peroxidation of the cell membrane and thus the loss of integrity [16]. In this experimental study, the possible hepatoprotective and cardioprotective effects of kefir, which is well known for its antioxidant and antitumor effects, against CPx-induced oxidative stress were investigated.

## 2. Materials and Methods

### 2.1. Kefir Fermentation

Commercially supplied and freeze-dried kefir yeast (Fairment Starter Kit—Bio kefir yeast—Berlin/Germany) and milk (Dost, 1 L golden full-fat pasteurized cow’s milk) were chosen. Three groups of kefirs were created, with fermentation at 24–26 °C at intervals of 24, 48, and 72 h. They were then kept at +4 °C, ready for use [2]. We gave kefirs to rats using gavage method for 12 days. 

### 2.2. Chemicals, Injections

Cyclophosphamide (CPx) (Sigma-Aldrich, St. Louis, MO, USA) was commercially available. A total of 500 mg of CPx was dissolved in 25 mL bidistilled water to prepare for injection of 150 mg/kg CPx. The injection was performed as a single dose intraperitoneally (i.p.)/bodyweight (b.w.) on the 12th day of the experiment, using sterile disposable syringes.

### 2.3. Experimental Design

In our experimental study, healthy, male Wistar albino rats were used which were 200 ± 20 gr and about 3 months old. During the experiment, the animals were kept in rooms with 12:12 light/dark cycle, humidity of 45–50% and a temperature of 22 ± 2 °C. And they were given tap water and normal pellet feed. The 42 rats used in this study were divided into 6 groups, with each group containing 7 rats. Group 1 was the control; Group 2 had single dose of 150 mg/kg/b.w CPx; Group 3 had 5 mg/kg/b.w kefir; Group 4 had 10 mg/kg/b.w kefir; Group 5 had 5 mg/kg/b.w kefir + 150 mg/kg/b.w CPx; and Group 6 had 10 mg/kg/b.w kefir + 150 mg/kg/b.w CPx. Fermented kefirs in specified doses were given to rats using gavage method for 12 days. A single dose of CPx was given i.p. on the last day of the experiment (the 12th day). At the end of the experiment, biochemical parameters were recorded, and cardiac and hepatic tissues were taken under anesthesia.

### 2.4. Biochemical Analyses

Alanine aminotransferase (ALT) (Cat no: OttoBC128), aspartate aminotransferase (AST) (cat no OttoBC127), alkaline phosphatase (ALP) (Cat no: OttoBC124), lactate dehydrogenase (LDH) (Cat no: OttoBC129) and creatinine kinase-MB (CK-MB) (cat no: OttoBC136) levels were measured using colorimetric method using Mindray-BS400 model fully automatic biochemistry device (Otto Scientific, Shanghai, China). Colorimetric assay was performed in accordance with the standardized method.

### 2.5. Troponin I (Trop I)

A sandwich ELISA was used. The micro-ELISA plate provided in the kit used is pre-coated with an antibody specific to rat cTnT/TNNT2. Samples (or standards) are added to the micro-ELISA plate wells and combined with the specific antibody. Then a biotinylated detection antibody specific for tat cTnT/TNNT2 and Avidin horseradish peroxidase (HRP) conjugate are added successively to each microplate well and incubated. Free components are washed away. The substrate solution is added to each well. Only the wells that contain rat cTnT/TNNT2, biotinylated detection antibody and Avidin HRP conjugate appear blue in color. The enzyme–substrate reaction is terminated by the addition of a stop solution, and the color becomes yellow. The optical density (OD) is measured spectrophotometrically at a wavelength of 450 nm ± 2 nm.

### 2.6. Ischemia Modified Albumin (IMA)

IMA was measured by colorimetric method using Mindray-BS400 device (RelAssay, Shanghai, China). The colorimetric assay format quantitatively measures unbound cobalt remaining after cobalt–albumin binding has occurred. Thus, with reduced cobalt–albumin binding, there is more free, unbound cobalt detected, resulting in elevated assay levels.

### 2.7. Histopathology

Prior to being examined under a light microscope, tissue samples were preserved in a 10% neutral buffer formaldehyde solution. Following identification, tissue samples were put into cassettes and rinsed for two hours under running water. Tissues were run through a succession of increasing alcohol concentrations (60, 70%, 80%, 90%, 96% and 100%) in order to extract water. The tissues were then polished by passing them through xylene before being implanted in melted paraffin. For each group, 4-micron-thick slices were cut from paraffin blocks and stained with hematoxylin and eosin stain. Using the Leica Q Vin 3 program on the Leica DCM 4000 computer-aided imaging system (Bensheim, Germany), the sections were assessed and captured on camera. A criteria table was created as a result of the evaluations made with hematoxylin and eosin staining.

### 2.8. Statistical Analysis

The quantitative values obtained at the end of the study were evaluated by applying the Duncan test after one-way ANOVA, which is used in the statistical analysis of more than two independent groups, using the SPSS 26.00 statistical data program.

## 3. Results

Since kefir creates different microbial flora, the fermented kefirs were tested on different days. Because no significant change was observed between the kefirs of the 1st, 2nd, and 3rd days, the kefirs of the three days were mixed and used. Studies show that kefirs are used in very different doses and for various durations. In this experimental study, we gave kefir to rats using the gavage method for 12 days, as in the studies of Cooper (1986) and Matsuu et al. (2003) [17,18].

In our study, the antioxidant and cell-protective effects of kefir on CPx-induced cardiotoxicity and hepatotoxicity were examined. Histopathological scoring according to hepatocyte degeneration, piecemeal necrosis, confluent necrosis, portal inflammation, fibrosis and spotty necrosis is shown in Table 1. According to this table, severe changes were observed in the CPx-administered group (a score of 3). Among the kefir + CPx groups, the poor condition in the fifth group approached that of the control (score 0) despite the CPx, while it was observed to improve with a slight change in the sixth group (a score of 1).

According to the scoring based on hyalinization, necrosis, dissociation and congestion in the rats’ cardiac muscle fibers, while there was a severe change (score 3) in the CPx-administered group, it resolved into a mild change (score 1) in the CPx + kefir-administered groups and approached that of the control group (Table 1).

When the CPx-treated groups’ liver and heart tissues were assessed, a statistically significant difference was seen when compared to the control group (Table 1). In contrast, it was shown that in the groups given varying dosages of kefir + CPx, the level of CP-induced tissue damage was reduced compared to that in the group given CP alone. In contrast, 10 mg/kg of kefir was more effective as a therapy in avoiding tissue damage due to CPx than 5 mg/kg of kefir.

In Figure 1, Figure 2, Figure 3, Figure 4, Figure 5, Figure 6 and Figure 7, the ALT, AST, ALP, LDH, IMA, CK-MB and Trop I levels, which are biochemical parameters indicative of cardiotoxicity and hepatotoxicity, were all found to be high in the second group given CPx. This increase was lower in the groups given kefir and kefir + CPx. According to our findings, these lower observed values due to kefir were mostly higher in the sixth group, which was given CPx + 10 mg/kg kefir (Figure 1, Figure 2, Figure 3, Figure 4, Figure 5, Figure 6 and Figure 7).

According to the rats’ cardiac tissue histopathology, degeneration (blue arrow) and necrosis (black arrow) are observed in the cardiac tissue of the rats treated with CPx (Figure 8b). The cardiac tissue of the rats treated with kefir has an appearance close to that of the control (Figure 8c,e). Pleomorphic nuclei (shown by the yellow arrow) and focal necrosis (shown by the black arrow) are observed in the cardiac tissue of the rats administered CPX + 5 mg/kg kefir (Figure 8d). A focal hyalinization area (gray arrow) is observed in the cardiac tissue of the rats administered CPx + 10 mg/kg kefir (Figure 8f).

According to the liver histopathology shown in Figure 9b, widespread necrosis (indicated by the black star) and parenchymal degeneration were observed in the liver parenchyma of the CPx-treated rats. The liver parenchyma and portal areas (indicated by the blue star) of the rats administered kefir are shown (Figure 9c,e). Areas of hemorrhaging (indicated by the black arrow) and degeneration were observed in the liver parenchyma of the rats administered CPx and 5 mg/kg kefir (Figure 9d). In Figure 9f, the liver parenchyma and portal area (indicated by the blue star) of the rats administered CPx + 10 mg/kg kefir are shown (Figure 9).

## 4. Discussion

The present study aims to assess how different kefir concentrations can lessen the adverse effects that the antineoplastic drug CPx has on the liver and heart. The main purpose of chemotherapy is to stop or destroy tumor growth without damaging healthy cells. However, since antineoplastic drugs have low selective properties, although they destroy cancer cells, they may also have some side effects on healthy cells. Studies are continuing to develop more effective methods that prevent the toxic effects of many antineoplastic drugs, such as cyclophosphamide (CPx), to allow them to be used safely in higher doses. In our experimental study, we aimed to examine the antioxidant and cytoprotective effects of kefir on the toxicity levels in the serum, heart, and liver tissue of CPx-treated rats. Since the stimulation of antioxidant defense mechanisms can reduce the biological effect of ROS, we used the natural fermented product kefir, which has antioxidant or anti-inflammatory effects as well as anti-cancer properties, to prevent oxidative damage by CPx. The chemical composition of kefir may vary depending on the fermentation time. However, according to the results of our previous study [2], it was determined that there was no significant difference in the content of kefir fermented for different periods of time. Therefore, we used kefirs with different fermentation durations and mixed them. Only the acidity rate changed, but acidity does not have a negative effect on nutritional value.

CPx, which has cytotoxic properties and is used in the treatment of various types of cancer, causes serious toxic side effects in tissues such as the heart and liver, so its use in high doses is limited. The liver is an organ that is highly susceptible to toxicity and damage due to its vital function in metabolizing drugs and toxins. According to Table 1, in which histopathological scores are given according to hepatocyte degeneration, piecemeal necrosis, confluent necrosis, portal inflammation, fibrosis and spotty necrosis, severe changes were observed in the CPx group (with a score of 3). Despite the CPx, the fifth group, to whom CPx + 5 mg/kg kefir was given, scored poorly, with levels approaching that of the control (with a score of 0), while in the sixth group, this severe change improved to a mild change (with a score of 1). In an experimental study, it was reported that liver tissue damage was caused by CPx metabolites [16]. According to the scores of hyalinization, necrosis, dissociation and congestion in the heart muscle fibers, a severe change was observed in the CPx group (with a score of 3), while in the CPx + kefir-administered groups, this severe change improved to a mild change (with a score of 1) and approached that of the control group (Table 1).

CPx, a nitrogen mustard alkylating oxazaphosphorine, has been used for over four decades in the treatment of numerous malignancies and immune-mediated illnesses [1]. Acrolein, phosphoramide and 4-hydroxy-CPx are produced when cytochrome P450 enzymes metabolically activate CPx, resulting in carcinogenic cell DNA damage. This is how CPx works to alkalinize the DNA of cancer cells. The use of this useful medication is unfortunately limited by the possibility that CPx will also damage typical human cells as a side effect. As a result of the toxic effects of CPx on the heart, kidney and liver, patients may experience severe morbidity or even death [3,9]. Other organs affected by CPx toxicity include the bone marrow, brain, lungs, gastrointestinal system, ovaries and testes [1,3,10]. As per prior research, oxidative stress is the primary factor responsible for tissue damage caused by CPx [4,10]. For instance, ALT, AST and LDH levels, which are markers of liver tissue damage, were found to rise dramatically following the administration of CPx (200 mg/kg) in experimental research [11]. According to Cengiz et al., CPx seriously harms liver tissue, and increases in ALT, AST and ALP levels in subject treated with CPx could indicate oxidative-stress-related damage [19]. In Shanholtz’s (2011) study, it was reported that high doses of CPx caused fatal cardiotoxicity [20]. Natural antioxidants such as kefir, which is a free radical scavenger and has been subject to experimental and clinical studies, are the most important agents needed to prevent oxidative stress and cytotoxicity caused by chemotherapeutic drugs. So the results showed that kefir reduced abnormal pathological symptoms, such as CPx-induced tissue damage, and heart and liver tissue was significantly improved.

The toxic effect of CPx occurs when its active metabolite, ACR, destroys antioxidant defense systems and creates high amounts of free radicals. When oxidative stress is stimulated, the balance between pro-oxidation and anti-oxidation, which is vital for the maintenance of homeostasis, turns in favor of pro-oxidation, leading to toxicity and therefore tissue and cell damage. In our study, the biochemical parameters ALT, AST, ALP and LDH, which are indicators of toxicity, were all found to be high in the second group given CPx, while this increase was less pronounced in the groups given kefir and CPx + kefir. The CPx-induced increase in our biochemical results is an indication of the deterioration of heart and liver tissue and therefore oxidative damage. Studies have also shown that CPx induction causes hepatoxicity by increasing ALT, AST, ALP and LDH levels [21,22]. In many other studies, it has been reported that CPx applications at various doses increase enzymes such as AST, ALT, ALP, CK and LDH and cause histopathological changes [23,24,25]. Some antioxidative agents are used to suppress the high-dose effects of antineoplastic drugs and reduce oxidative stress. According to our findings, the decrease in the groups given kefir + CPx was the most pronounced in the sixth group, which was given CPx + 10 mg/kg kefir, despite the CPx (Figure 1, Figure 2, Figure 3, Figure 4, Figure 5, Figure 6 and Figure 7).

In our study, our liver histopathology findings showed widespread necrosis (indicated by the black star) and parenchymal degeneration liver damage in the liver parenchyma of the CPx-treated rats (Figure 9b). In the liver parenchyma of the CPx + 5 mg/kg kefir-administered group, areas of bleeding (indicated by the black arrow) and degeneration were observed within the parenchyma (Figure 9d). The liver parenchyma and portal area (indicated by the blue star) of the rats administered CPx +10 mg/kg kefir are shown (Figure 9f). The histopathological findings described in our study confirm that the toxic effect of CPx does not destroy the antioxidant defense systems of acreolin and causes high amounts of free radical formation. In this context, kefir, which has strong antioxidant properties, was used in our study to allow the use of CPx in higher doses, and it was found to be effective. In studies, it has been reported that kefir consumption has antioxidant and anticarcinogenic effects, in parallel with the results of our study [2,26]. Kefir also provides antitumor and immunotherapeutic effects. Immunotherapeutic methods developed to prevent the growth of tumors and to treat cancer by stimulating the immune response against tumor cells are of great importance in the treatment of cancer [27].

CPx is known to cause fatal cardiotoxicity. CK-MB, IMA, AST and Trop I levels, which are biomarkers of cardiac damage, were found to be high in the CPx groups of our study. This shows that cardiac tissue is damaged in the presence of CPx. In a study, it was reported that cardiac biochemical markers, such as CK-MB, LDH and IMA, which are considered signs of cellular changes in the myocardium, increased significantly in the CPx group [28,29]. Likewise, an experimental study using rats reported that 200 mg/kg CPx caused oxidative stress in the heart tissue, as well as an increase in CK-MB and LDH levels and myocardial dysfunction [30]. Another study emphasized that CPx increased serum LDH and CK-MB levels in cardiac tissue, and acute heart failure and histopathological lesions were observed [31]. There is also another study in which Trop I and CK-MB expression was found to be significant between a control group and CPx group [32]. Despite the CPx toxicity in the kefir-administered groups, the CK-MB, IMA, AST and Trop I levels decreased and approached those of the control; thus, the kefir appeared to have a protective effect. The histopathology of heart tissue in our study is compatible with this biochemical picture. Degeneration (indicated by the blue arrow) and necrosis (indicated by the black arrow) are observed in the heart muscle of rats treated with CPx (Figure 8b). While pleomorphic nuclei (indicated by the yellow arrow) and focal necrosis (indicated by the black arrow) are observed in the heart muscle of the rats administered CPx + 5 mg/kg kefir (Figure 8d), a focal hyalinization area (indicated by the gray arrow) is observed in the heart muscle of the rats administered CPx + 10 mg/kg kefir (Figure 8f). Also, studies have shown that in the cardiac tissue of experimental groups given CPx, there are changes in the staining properties of the cell cytoplasm, a darker color and shrinkage in the nuclei, bleeding between muscle cells, small inflammatory cell foci and the separation of muscle fibers [12,25]. The histological findings of the experimental study conducted by Motawi et al. (2010) involving the administration of 200 mg/kg CPx showed that there were hemorrhagic foci in the myocardium, deterioration and hyalinization in the myocardial fibers [33]. As can be seen, the histological findings of our study are compatible with the findings of other researchers. In this study, it was seen that kefir was effective against the toxic effects of CPx on the liver and heart tissue in cases in which high doses were used. All these findings show that CPx metabolites damage the heart and liver tissue membranes, and biochemical studies support this result. This research showed that, to some extent, kefir has protective effects on CPx-induced cardiotoxicity and hepatotoxicity in rats. Most approved drugs for cancer treatment show drawbacks such as low potency, drug resistance, and/or toxicity. For such reasons, researchers constantly continue to develop and discover new agents with ideal properties [34].

## 5. Conclusions

Studies are still continuing to develop methods that prevent the toxic effects of many antineoplastic chemical agents, especially CP, and allow them to be used in higher doses and for longer periods of time. When these drugs are used in high doses and frequently, the desired treatment effects disappear, and they cause harmful side effects. As a matter of fact, in our study, the most severe pathological findings were observed only in the CPx group, and it was concluded that CPx had a very toxic effect on the heart and liver. Kefir is a very effective agent, reducing the side effects of CPx as a cancer immunotherapy that uses the power of the patient’s own immune system to destroy cancerous cells. As a result, it has been observed that kefir reduces tissue damage by suppressing oxidative stress and has a cell-protective effect against CPx-induced hepatotoxicity and cardiotoxicity, and it has been concluded that more comprehensive and advanced studies are needed to elucidate the role of kefir in oxidative damage that occurs with the chemotherapeutic response. In addition, we hope to contribute to the literature by conducting scientific studies in the future to determine if there are more effective doses of and times of administration for kefir.

## Figures and Tables

**Figure 1 biomedicines-12-01199-f001:**
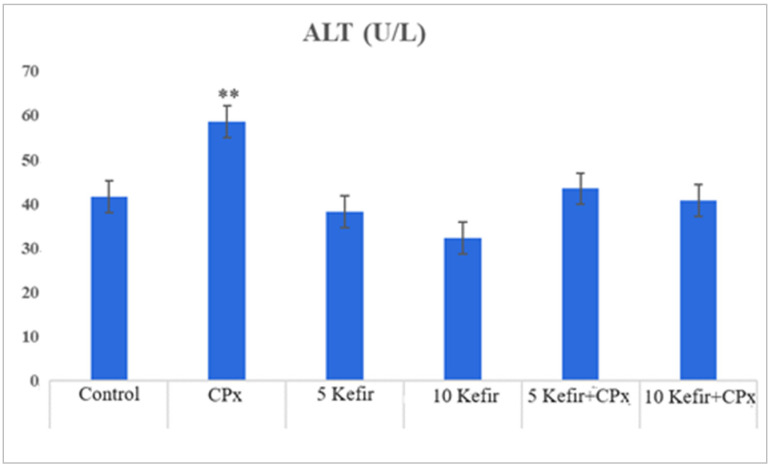
ALT values of experimental groups (** *p* < 0.05 compared to control).

**Figure 2 biomedicines-12-01199-f002:**
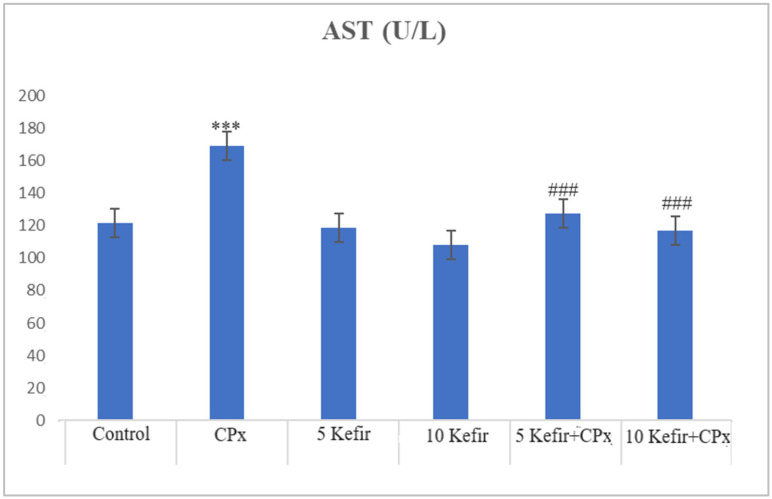
AST values of experimental groups (*** *p* < 0.001 compared to control; ^###^
*p* < 0.001 compared to CPx group).

**Figure 3 biomedicines-12-01199-f003:**
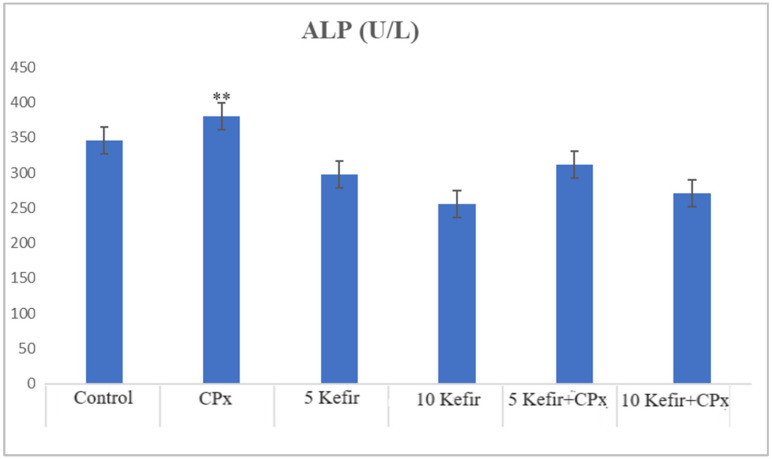
ALP values of experimental groups (** *p* < 0.05 compared to control).

**Figure 4 biomedicines-12-01199-f004:**
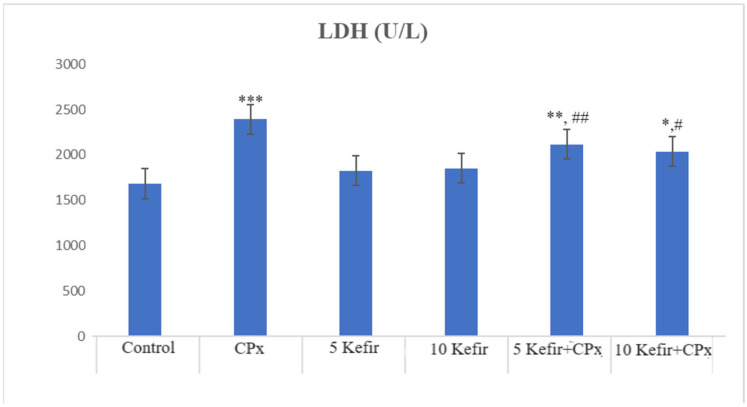
LDH values of experimental groups (*** *p* < 0.001 compared to control; ** *p* < 0.05 compared to control; * *p* < 0.01 compared to control; ^##^
*p* < 0.05 compared to CPx group; ^#^
*p* < 0.01 compared to CPx group).

**Figure 5 biomedicines-12-01199-f005:**
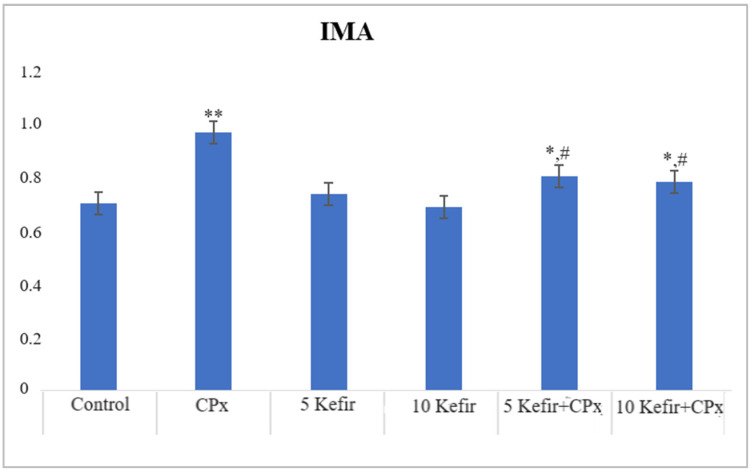
IMA (g/dL) values of experimental groups (** *p* < 0.05 compared to control; * *p* < 0.01 compared to control; ^#^
*p* < 0.01 compared to CPx group).

**Figure 6 biomedicines-12-01199-f006:**
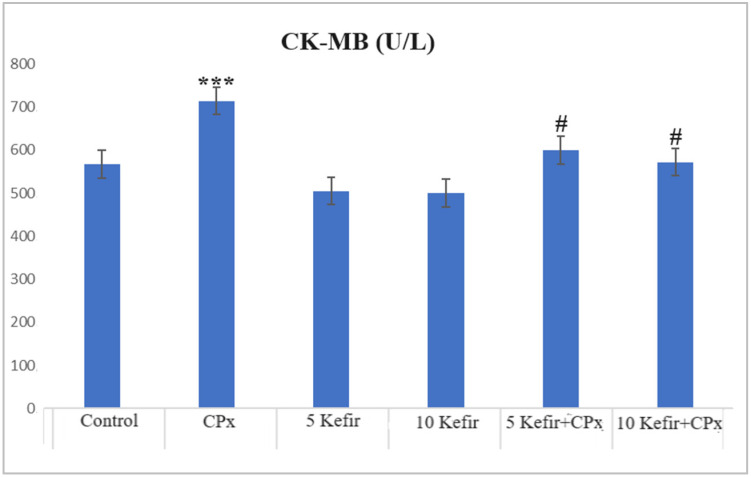
CK-MB values of experimental groups (*** *p* < 0.001 compared to control; ^#^
*p* < 0.01 compared to CPx group).

**Figure 7 biomedicines-12-01199-f007:**
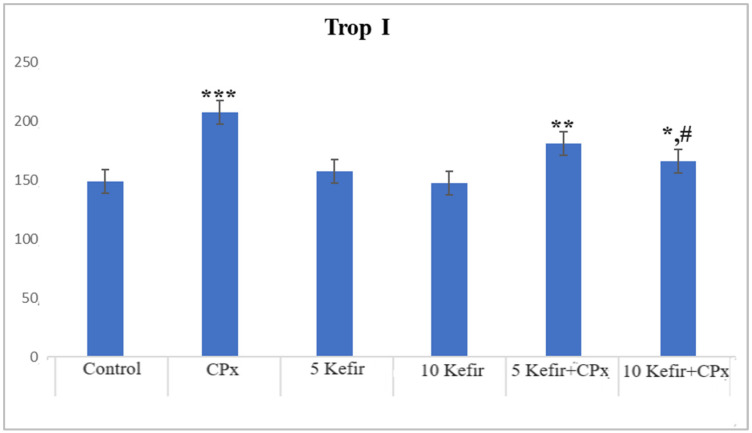
Trop I values of experimental groups (*** *p* < 0.001 compared to control; ** *p* < 0.05 compared to control; * *p* < 0.01 compared to control; ^#^
*p* < 0.01 compared to CPx group).

**Figure 8 biomedicines-12-01199-f008:**
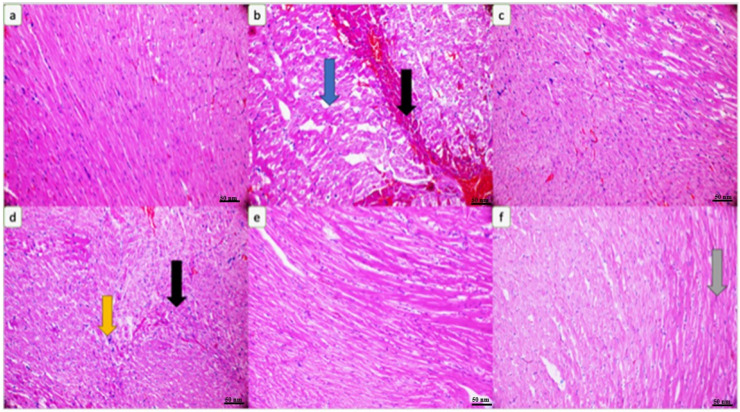
(**a**) Cardiac tissue of rats in the control group. (**b**) Degeneration (blue arrow) and necrosis (black arrow) in the cardiac tissue of rats treated with CPx. (**c**) Cardiac tissue of rats treated with kefir. (**d**) Pleomorphic nuclei (yellow arrow) and focal necrosis (black arrow) in the cardiac tissue of rats treated with CPx and kefir. (**e**) Cardiac tissue of rats treated with kefir. (**f**) Focal hyalinization area in the cardiac tissue of rats treated with CPx and kefir (gray arrow) (H&E, ×200).

**Figure 9 biomedicines-12-01199-f009:**
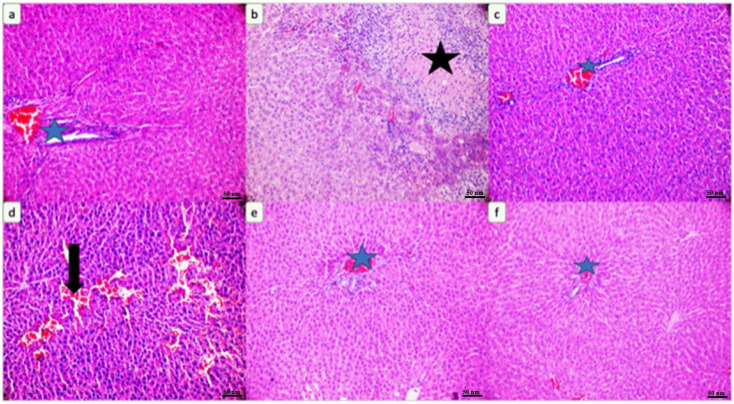
(**a**) Liver parenchyma and portal areas (blue star) of rats in the control group. (**b**) Widespread necrosis (black star) and parenchymal degeneration in the liver parenchyma of CPx-treated rats. (**c**) Liver parenchyma and portal areas (blue star) of kefir-treated rats. (**d**) Areas of hemorrhaging (black arrow) and degeneration in the liver parenchyma of rats treated with CPx and kefir. (**e**) Liver parenchyma and portal areas (blue star) of kefir-treated rats. (**f**) Liver parenchyma and portal area (blue star) of rats administered CPx and kefir (H&E, ×200).

**Table 1 biomedicines-12-01199-t001:** Histopathological scoring of control, 150 mg/kg CPx, 5 and 10 mg/kg kefir and 5 and 10 mg/kg kefir + 150 CPx applied experimental groups, according to damage and inflammation status.

Groups	Cardiac Score	Hepatic Score
Control	0	0
150 mg/kg CPx	3 ^a^	3 ^a^
5 mg/kg kefir	0	0
l0 mg/kg kefir	0	0
5 mg/kg kefir +150 CPx	2 ^a^	2 ^a^
10 mg/kg kefir +150 CPx	1 ^b,c^	1 ^b,c^

0: No change, 1: slight change, 2: moderate change, 3: severe change. ^a^
*p* < 0.001 (compared to the control group); ^b^
*p* < 0.01 (compared to 150 mg/kg CPx group); ^c^
*p* < 0.05 (compared to 5 mg/kg kefir +150 CPx).

## Data Availability

Data are contained within the article.

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
