# Peer review of "In Vitro Antitumor and Antioxidant Capacity as well as Ameliorative Effects of Fermented Kefir on Cyclophosphamide-Induced Toxicity on Cardiac and Hepatic Tissues in Rats"

_biomedicines, 2024, doi:10.3390/biomedicines12061199_

Round 1

Reviewer 1 Report

Comments and Suggestions for Authors

The manuscript presents several studies in order to demonstrate the beneficial effect of kefir in ameliorating the toxic effects of cyclophosphamide administration. In my opinion, in order to be published in this journal, the manuscript needs a major revision, which requires completion with additional studies to demonstrate these properties for kefir. The answers to the following questions should also be found in it:

1. What were the reasons why kefir was chosen to counteract the toxic effects of cyclophosphamide over other substances with the same properties? 

2. Why was its effect on the cardiac toxicity of cyclophosphamide studied considering that it is not one of the main adverse effects following the administration of this drug?

3. Studies to demonstrate the antitumor effect of kefir in vitro and apoptosis studies must be completed with studies on cell cultures, with results that confirm these properties of kefir.

Author Response

Response Reviwer 1

  1. Cyclophosphamide (CTX) is a cytotoxic drug and widely preferred in cancer therapy but can suppress both humoral and cellular immunity, has an immunosuppressive effect, causes multiple organ toxicity such as the testicular toxicity, and the use of effective high doses is restricted. That's why we chose kefir not only for its antioxidant and antitumor effects, but also especially for its immunotherapeutic effect.

  1. In our entire study, the multi-organ toxicity of cyclophosphamide was examined. In this manuscript, only the cardiac and hepatic damage of cyclophosphamide was reported.

  1. Thank you for your advice and suggestions. We will also carry out such researc in the future.

Reviewer 2 Report

Comments and Suggestions for Authors

Dear authors,

The work with title “In vitro antitumor, antioxidant capacity and ameliorative effects of fermented kefir on cyclophosphamide-induced toxicity on cardiac and hepatic tissues in rats” represents a valuable contribution regarding Fermented prebiotic and probiotic products kefir biological effects which could prevent the growth of tumors by stimulating the immune response against tumor cells. The using of cyclophosphamide which is widely preferred in cancer therapy but its uses was is restricted because of it is disadvantages. In this work 42 rats were used and divided to groups it was observed that kefir had a cytoprotective effect against CPx-induced oxidative stress, hepatotoxicity and cardiotoxicity, bringing biochemical parameters closer to control by suppressing oxidative stress and reducing tissue damage. All of these findings seem valuable and suitable contribution to be published in the biomedicines Journal after justifying the following points:

·       In the abstract line 16 phrase “causes toxicities” should be changed to “side effects”, since this drug’s main action is the cytotoxicity

·       The groups naming should be edited again for example the first group called control group but the second one also control, so may you can edit these groups as Latin numbers, and rename them accordingly in the whole manuscript

·       It is very important to identify the first group as negative control and the second group as positive control.

·       It is recommended to remove any abbreviation from the abstract like i.p.

·       The Introduction section is too short, since the main goal of this study was to improve or remove the side effects of an anticancer drug and the relation of oxidative stress these two main point cancer, and oxidative stress should be highlighted by two paragraphs, the following references could be used accordingly, for cancer and anticancer effects use “ Synthesis, chemo-informatics, and anticancer evaluation of fluorophenyl-isoxazole derivatives” + “ Design and synthesis of novel substituted indole-acrylamide derivatives and evaluation of their anti-cancer activity as potential tubulin-targeting agents” and “Biomolecules 2022, 12, 1843.” That will make the frequency of the story more attractive.

·       For the oxidative stress and antioxidant you can use “In vitro and in vivo assessment of the antioxidant potential of isoxazole derivatives” and “Free radicals and enzymes inhibitory potentials of the traditional medicinal plant Echium angustifolium” which will improve this section too

·       The proposed mechanism of action of kefir should be discussed accordingly

·       The Aim of this study lines 56-58 should be improved too

·       Each protocol in the method section should be cited accordingly

·       Line 71 control and edit the control group as negative control, positive control an go on the other groups.

·       Line 87- 88 was written in Turkish language!

·       Use the same font in figures and text

·       What further preclinical or clinical studies are needed to validate the therapeutic potential of this experiments work, particularly considering its promising effects observed in this study?

Best wishes

Author Response

Response to Reviewer 2

·       In the abstract line 16 phrase “causes toxicities” should be changed to “side effects”, since this drug’s main action is the cytotoxicity. : corrected as requested.

·       The groups naming should be edited again for example the first group called control group but the second one also control, so may you can edit these groups as Latin numbers, and rename them accordingly in the whole manuscript. : corrected as requested.

·       It is very important to identify the first group as negative control and the second group as positive control. : We named the first group (Group 1) as control and the second group (Group 2)  as 150 mg/kg CPx. It was like this in the entire literature review and the first group is called control.

·       It is recommended to remove any abbreviation from the abstract like i.p. : corrected as requested.

·       The Introduction section is too short, since the main goal of this study was to improve or remove the side effects of an anticancer drug and the relation of oxidative stress these two main point cancer, and oxidative stress should be highlighted by two paragraphs, the following references could be used accordingly, for cancer and anticancer effects use “ Synthesis, chemo-informatics, and anticancer evaluation of fluorophenyl-isoxazole derivatives” + “ Design and synthesis of novel substituted indole-acrylamide derivatives and evaluation of their anti-cancer activity as potential tubulin-targeting agents” and “Biomolecules 2022, 12, 1843.” That will make the frequency of the story more attractive.   : supported by recommended manuscripts

·       For the oxidative stress and antioxidant you can use “In vitro and in vivo assessment of the antioxidant potential of isoxazole derivatives” and “Free radicals and enzymes inhibitory potentials of the traditional medicinal plant Echium angustifolium” which will improve this section too. : Thank you for your suggested studies, I scanned them all.

·       The proposed mechanism of action of kefir should be discussed accordingly.  Improved

·       The Aim of this study lines 56-58 should be improved too Improved.

·       Each protocol in the method section should be cited accordingly : Improved

·       Line 71 control and edit the control group as negative control, positive control an go on the other groups. : edited

·       Line 87- 88 was written in Turkish language! : corrected

·       Use the same font in figures and text : edited

·       What further preclinical or clinical studies are needed to validate the therapeutic potential of this experiments work, particularly considering its promising effects observed in this study? : This explanation was developed under the title of concllucision.

Round 2

Reviewer 1 Report

Comments and Suggestions for Authors

 The authors have improved the article according to the suggestions provided

Author Response

Thank you very much for considering the article. I would also like to thank you for your valuable suggestions, contributions and time.

Reviewer 2 Report

Comments and Suggestions for Authors

The authors should discuss the results of the statistical analysis, was these results significant according to each group?

The authors MUST include correlations between all activities tested, by using Pearson correlation analysis to do that.  

You should take you efforts to improve the introduction by the important recent related works which was not resolved, and mentioned before to support the frequency story of this manuscript

Some figures font still need improvement

The discussion section is very weak, this work should be discussed very well with the other related recent work  

Best wishes  

Author Response

Response to Reviewers

We would like to thank the reviewers for their careful and thorough reading of this manuscript, apart from their thought-provoking comments and constructive suggestions, which have really helped improve the quality of this manuscript. All the necessary corrections have been made in the light of the suggestions of both referees as presented below.

We would like to thank you for your time and interest in our manuscript, which hopefully is now ready for publication in your journal. Look forward to hearing from you.

Sincerely yours,

Asst. Prof. Songül ÇETÄ°K YILDIZ

Please find our responses below with the reviewer’s comments highlighted in italics.

Response to Reviewer 1

We whole-heartedly appreciate the positive feedback from the reviewer.

  1. The authors should discuss the results of the statistical analysis, was these results significant according to each group?

Reply: the results of the statistical analysis discussed. All groups were compared statistically, and the results of the groups with statistical significance are presented in Table 1.

  1. The authors MUST include correlations between all activities tested, by using Pearson correlation analysis to do that.

Reply: The suggested correlation test results and obtained statistical significances between all tested activities have been added in to the manuscript.

  1. You should take you efforts to improve the introduction by the important recent related works which was not resolved, and mentioned before to support the frequency story of this manuscript.

Reply: Done.

  1. Some figures font still need improvement

Reply: Corrected.

  1. The discussion section is very weak, this work should be discussed very well with the other related recent work .

Reply: Corrected.

Round 3

Reviewer 2 Report

Comments and Suggestions for Authors

The authors were improved the manuscript accordingly 

Author Response

Thank you for your evaluations.